# Ni, Co and Ni-Co-Modified Tungsten Carbides Obtained by an Electric Arc Method as Dry Reforming Catalysts

Zhanar Bolatova [1], Dmitrii German [2], Ekaterina Pakrieva [2], Alexander Pak [1], Kirill Larionov [1], Sónia A. C. Carabineiro [3], Nina Bogdanchikova [4], Ekaterina Kolobova [2,*] and Alexey Pestryakov [2,*]

1. School of Energy & Power Engineering, National Research Tomsk Polytechnic University, Lenin Av. 30, 634050 Tomsk, Russia
2. Research School of Chemistry & Applied Biomedical Sciences, National Research Tomsk Polytechnic University, Lenin Av. 30, 634050 Tomsk, Russia
3. LAQV-REQUIMTE, Department of Chemistry, NOVA School of Science and Technology, Universidade NOVA de Lisboa, 2829-516 Caparica, Portugal
4. Centro de Nanociencias y Nanotecnología, Universidad Nacional Autónoma de México, Ensenada 22800, Mexico
* Correspondence: ekaterina_kolobova@mail.ru (E.K.); pestryakov2005@yandex.ru (A.P.)

**Abstract:** Dry reforming of methane (DRM), to produce synthesis gas, is one of the most important chemical reactions used for the industrial production of hydrogen and leads to the synthesis of hydrocarbons (liquid fuels) and other valuable products. A cost-effective alternative to active and stable noble metal DRM catalysts, with comparable catalytic performance, can be composite materials based on nickel, cobalt and transition metal carbides. In this line, the present work proposes a non-standard way to obtain dry reforming catalysts of Ni, Co and Ni-Co-modified tungsten carbide (WC) produced by an electric arc method. Different amounts of nickel, cobalt and their mixtures were deposited on tungsten carbide by deposition-precipitation with NaOH (DP) and incipient wetness impregnation (IWI) methods. The resulting materials were characterized by $N_2$ adsorption-desorption, transmission electron microscopy, energy dispersive spectroscopy, X-ray diffraction and X-ray photoelectron spectroscopy, and their performance was evaluated in DRM. The composition and preparation method of catalysts predetermined their structural, textural and electronic properties, playing a decisive role in their activity for DRM. DP-prepared 20%Ni/WC material remained resistant to oxidation, both that of the active metal (nickel) and of the tungsten carbide, as well as to coking during DRM. This sample proved to be the most active and stable among all studied materials. Possibly, the resistance to oxidation and coking was due to a more efficient implementation of the oxidation/(re)carbonization cycle on the surface of this catalyst.

**Keywords:** methane dry reforming; tungsten carbide; Ni- and Co-based catalyst; catalyst stability; oxidation; coking resistance

## 1. Introduction

The unquestionable importance of dry reforming of methane (DRM, Equation (1)) lies in the production of synthesis gas (also called syngas) with a $H_2/CO$ ratio near 1.0. One advantage of this process is the valorization of greenhouse gases, such as methane, a component of natural gas and biogas, and carbon dioxide, one of the main gaseous wastes. Syngas can be used for the Fischer–Tropsch process in the reaction of olefins' hydroxylation and in the industrial production of dimethyl ether [1]. The latter can be considered as a promising alternative to gasoline and diesel fuels, as well as a raw material for the production of other high-octane additives to gasoline, and oxygenates [2].

$$CO_2 + CH_4 \rightarrow 2CO + 2H_2 \tag{1}$$

However, the practical applications of DRM are hindered by the endothermicity of this high temperature reaction (+247 kJ) and inert reaction kinetics of methane and carbon dioxide molecules. Furthermore, the reaction is inevitably accompanied by carbon deposition on the catalyst surface due to side reactions occurring on the surface (Equation (2)), which lead to deactivation of the catalyst [3].

$$CH_4 \rightarrow C + 2H_2 \tag{2}$$

Many noble metal catalysts, such as Rh, Ru, Pd, Pt and Ir, show high catalytic performance under high-temperature DRM process conditions, with no significant amount of carbon deposition, due to the lower solubility of carbon on the catalyst surface [4–6]. However, low availability and a high cost of noble metals greatly limit their industrial application.

Alternative inexpensive supported Ni and Co metal catalysts have been used due to their high activity, comparable to the activity of noble metals [7–10]. However, the main disadvantages of Ni and Co-based catalysts are deactivation due to coke deposition and active metal sintering at high reaction temperatures. The type and nature of carbon deposits are highly dependent on the active metal, support and the reaction temperature. Therefore, active developments are currently underway to modify the supports of Ni and Co catalysts.

Recently, transition metal carbides (TMCs), especially tungsten and molybdenum, aroused notable interest due to their excellent catalytic activity, thermal stability and selectivity for a wide range of reactions, such as hydrocarbon reforming, hydrogenation and CO oxidation [11–13]. In addition, the similarity of the electronic structure of carbides and noble metals [14,15] turns carbide catalysts into desirable alternatives to noble metal catalysts for DRM [16–19]. Under DRM conditions and elevated pressures, tungsten and molybdenum carbides are not prone to sintering, coking or poisoning with sulfur compounds, unlike nickel and cobalt catalysts [20]. However, under lower pressures, TMCs exhibit poor oxidation stability, resulting in catalyst deactivation by conversion of these active carbides into inefficient oxides, as explained by Thomson [21].

Therefore, the combination of a highly active and inexpensive component (Ni and Co) with stable compounds (WC, $Mo_2C$) seems a promising alternative in the development of new catalysts with higher activity and resistance to oxidation and poisoning by carbon deposits in the dry reforming process [22–26].

This work aims at showing one of the non-standard ways of obtaining dry reforming catalysts based on tungsten carbide (WC) synthesized by a vacuum-free electric arc method, and modified with different amounts of nickel, cobalt and their mixtures by deposition-precipitation with NaOH (DP) and incipient wetness impregnation (IWI), and also to evaluate the effect of the preparation method and catalyst composition on its physico-chemical and catalytic properties for DRM.

## 2. Results and Discussion

### 2.1. Characterization of Support and Catalysts

According to X-ray powder diffraction data, the support obtained by a vacuum-free electric arc method consists of hexagonal phases of α-WC (ICDD no. 00-025-1047) and β-$W_2C$ (ICDD no. 00-035-0776) (Table 1, entry 1, Figure S1).

The XRD patterns of catalysts obtained by the incipient wetness impregnation (IWI) method show characteristic peaks corresponding to the monoclinic phase of $WO_2$ (Table 1, entries 2, 4 and 6, Figure S1), the orthorhombic phase of $WO_3$ (Table 1, entries 8, 10, 12 and 14, Figure S1) and an insignificant number of initial phases of α-WC (Table 1, entries 2, 6, 8, 12 and 14, Figure S1) and β-$W_2C$ (Table 1, entry 2, Figure S1). The formation of tungsten oxides from its carbides is probably due to the oxidation of the latter by oxygen (air, water, oxygen released during the thermal decomposition of nickel and/or cobalt precursors) during the catalyst synthesis. In addition, for samples with more than 10 wt.% nickel and less than 10 wt.% cobalt, the characteristic reflections of the cubic phases of nickel and cobalt are observed (Table 1, entries 2, 4 and 6, Figure S1), while an increase in the cobalt content from 10 to 20 wt.%, with a simultaneous decrease in the nickel

amount from 10 to 1 wt.%, leads to the formation of monoclinic phases of cobalt and nickel tungstates (Table 1, entries 8, 10, 12 and 14, Figure S1). The formation of metal tungstates can occur as a result of a solid-phase reaction between oxides of tungsten and nickel and/or cobalt during calcination and/or pretreatment [27], or as a result of the oxidation of metal-tungsten carbides by a reaction occurring from 500 to 850 °C [28]: $3MWC + 8O_2 \rightarrow 3MWO_4 + 2CO + CO_2$, where M is nickel or cobalt. Regarding the reason why separate cubic metal phases are formed in the case of samples dominated by nickel (samples containing 15–20 wt.% Ni and 0–5 wt.% Co), while with a prevalence of cobalt (samples containing 10–20% wt.% Co and 0–10 wt.% Ni), nickel and cobalt tungstates are formed: It can be assumed that it is due to different decomposition temperatures of nickel and cobalt precursors ($Ni(NO_3)_2 \cdot 6H_2O$ and $Co(NO_3)_2 \cdot 6H_2O$) [29]; the latter decomposes at a lower temperature (has a different oxygen release rate), and the oxides have different reduction temperatures (cobalt oxide is reduced at a higher temperature than nickel oxide).

**Table 1.** Structural and textural characteristics of the studied catalysts and support.

| Entry | Sample | Observed Phase (Framework) | $w_i/\Sigma_w$ (%) | $R_{exp}$ (%) | SSA ($m^2$/g) | Pore Size (nm) | Pore Volume ($cm^3$/g) |
|---|---|---|---|---|---|---|---|
| 1 | WC | WC (hexagonal)<br>$W_2C$ (hexagonal) | 71<br>29 | 3.2 | 4.5 | 12.3 | 0.01 |
| 2 | 20%Ni/WC_IWI | Ni (cubic)<br>WC (hexagonal)<br>$W_2C$ (hexagonal)<br>$WO_2$ (monoclinic) | 27<br>9<br>7<br>57 | 2.1 | 15.6 | 17.3 | 0.08 |
| 3 | 20%Ni/WC_DP | Ni (cubic)<br>WC (hexagonal)<br>$W_2C$ (hexagonal) | 27<br>63<br>10 | 3.3 | 38.4 | 9.9 | 0.05 |
| 4 | 19%Ni_1%Co/WC_IWI | Ni (cubic)<br>Co (cubic)<br>$WO_2$ (monoclinic) | 13<br>traces<br>86 | 2.6 | 11.6 | 14.4 | 0.08 |
| 5 | 19%Ni_1%Co/WC_DP | Ni (cubic)<br>WC (hexagonal)<br>$W_2C$ (hexagonal) | 26<br>60<br>14 | 1.6 | 32.4 | 11.3 | 0.04 |
| 6 | 15%Ni_5%Co/WC_IWI | Ni (cubic)<br>Co (cubic)<br>WC (hexagonal)<br>$WO_2$ (monoclinic) | 16<br>13<br>1<br>70 | 1.8 | 10.8 | 14.7 | 0.09 |
| 7 | 15%Ni_5%Co/WC_DP | Ni (cubic)<br>Co (cubic)<br>WC (hexagonal)<br>$W_2C$ (hexagonal) | 22<br>6<br>61<br>11 | 2.3 | 37.8 | 12.2 | 0.05 |
| 8 | 10%Ni_10%Co/WC_IWI | $CoWO_4$ (monoclinic)<br>$NiWO_4$ (monoclinic)<br>WC (hexagonal)<br>$WO_3$ (orthorhombic) | 42<br>41<br>traces<br>17 | 1.5 | 13.6 | 19.7 | 0.07 |
| 9 | 10%Ni_10%Co/WC_DP | Ni (cubic)<br>Co (cubic)<br>WC (hexagonal)<br>$W_2C$ (hexagonal) | 16<br>11<br>62<br>11 | 1.7 | 55.7 | 4.4 | 0.06 |
| 10 | 5%Ni_15%Co/WC_IWI | $CoWO_4$ (monoclinic)<br>$NiWO_4$ (monoclinic)<br>$WO_3$ (orthorhombic) | 58<br>22<br>20 | 1.5 | 12.9 | 19.7 | 0.06 |

**Table 1.** *Cont.*

| Entry | Sample | Observed Phase (Framework) | $w_i/\Sigma_w$ (%) | $R_{exp}$ (%) | SSA ($m^2/g$) | Pore Size (nm) | Pore Volume ($cm^3/g$) |
|---|---|---|---|---|---|---|---|
| 11 | 5%Ni_15%Co/WC_DP | Ni (cubic)<br>Co (cubic)<br>WC (hexagonal)<br>$W_2C$ (hexagonal) | traces<br>traces<br>83<br>17 | 1.5 | 54.4 | 6.4 | 0.09 |
| 12 | 1%Ni_19%Co/WC_IWI | $CoWO_4$ (monoclinic)<br>$NiWO_4$ (monoclinic)<br>WC (hexagonal)<br>$WO_3$ (orthorhombic) | 46<br>traces<br>8<br>46 | 1.7 | 14.5 | 18.9 | 0.07 |
| 13 | 1%Ni_19%Co/WC_DP | Co (cubic)<br>Ni (cubic)<br>WC (hexagonal)<br>$W_2C$ (hexagonal) | 11<br>traces<br>72<br>17 | 2.7 | 40.5 | 11.0 | 0.11 |
| 14 | 20%Co/WC_IWI | $CoWO_4$ (monoclinic)<br>WC (hexagonal)<br>$WO_3$ (orthorhombic) | 30<br>4<br>66 | 1.6 | 13.6 | 16.4 | 0.06 |
| 15 | 20%Co/WC_DP | Co (cubic)<br>WC (hexagonal)<br>$W_2C$ (hexagonal) | 10<br>73<br>17 | 2.8 | 37.2 | 13.1 | 0.12 |
| 16 | WC_SP | WC (hexagonal)<br>$W_2C$ (hexagonal)<br>C (hexagonal) | 82<br>12<br>6 | 2.2 | - | - | - |
| 17 | 20%Ni/WC_IWI_SP | Ni (cubic)<br>WC (hexagonal)<br>$WO_2$ (monoclinic)<br>C (rhombohedral) | 15<br>48<br>19<br>18 | 3.9 | - | - | - |
| 18 | 20%Ni/WC_DP_SP | Ni (cubic)<br>WC (hexagonal)<br>$W_2C$ (hexagonal)<br>C (hexagonal) | 32<br>58<br>3<br>7 | 3.5 | - | - | - |
| 19 | 10%Ni_10%Co/WC_IWI_SP | $CoWO_4$ (monoclinic)<br>NiO (cubic)<br>WC (hexagonal)<br>$WO_2$ (monoclinic)<br>C (rhombohedral) | 12<br>16<br>2<br>52<br>18 | 3.3 | - | - | - |
| 20 | 10%Ni_10%Co/WC_DP_SP | $CoWO_4$ (monoclinic)<br>NiO (cubic)<br>WC (hexagonal)<br>$W_2C$ (hexagonal)<br>$WO_2$ (monoclinic)<br>C (rhombohedral) | 6<br>31<br>41<br>7<br>5<br>10 | 2.3 | - | - | - |
| 21 | 20%Co/WC_IWI_SP | $CoWO_4$ (monoclinic)<br>C (hexagonal) | 93<br>7 | 2.8 | - | - | - |
| 22 | 20%Co/WC_DP_SP | $CoWO_4$ (monoclinic)<br>WC (hexagonal)<br>$W_2C$ (hexagonal)<br>C (rhombohedral) | 50<br>33<br>3<br>14 | 1.6 | - | - | - |

SP—spent (used) support or catalyst. $w_i/\Sigma_w$—obtained phase weight proportions (%). $R_{exp}$ describes the quality of XRD fit (%), where a smaller value of $R_{exp}$ indicates a better fit to the measured data. SSA—specific surface area ($m^2/g$).

The samples synthesized by deposition-precipitation with NaOH (DP) show peaks related to phases of α-WC and β-$W_2C$, as well as peaks corresponding to the cubic phase

of Ni and/or Co (Table 1, entries 3, 5, 7, 9, 11, 13 and 15, Figure S1). In the case of DP, before the vacuum thermal treatment, nickel and cobalt are present on the support surface in the form of hydroxides, rather than nitrates, as in the case of IWI. Further, under the action of temperature and a vacuum, their hydroxides decompose to form the corresponding oxides and water. Due to the presence of a vacuum, the absence of free released oxygen and a lower drying temperature, tungsten carbide does not oxidize, and the subsequent reduction treatment leads to the formation of metallic nickel and cobalt on its surface.

Table 1 also presents the XRD results of the spent support and catalysts (Table 1, entries 16–22). XRD patterns of the tungsten carbide spent in DRM show a partial transition of the $\beta$-$W_2C$ phase to $\alpha$-WC phase, resulting from the carbonization process, as well as reflections related to hexagonal graphite (Table 1, entry 16, Figure S2). An analysis of XRD patterns of the spent catalysts also shows that phase transformations occur during the DRM reaction, and that reflections related to graphite in the rhombohedral or hexagonal allotropic modification appear in all used catalysts (Table 1, entries 17–22, Figure S2). It should be noted that rhombohedral graphite is a thermodynamically unstable allotropic form of graphite that gradually transforms into a hexagonal form upon heating [30]. The graphite content of all spent catalysts prepared by DP is lower than the values for the corresponding catalysts prepared by IWI, except for the material containing 20 wt.% of Co (Table 1, entry 22). Samples containing 20 wt.% of Ni were the most stable in DRM (Table 1, entries 17 and 18, Figure S2). The phase transformation of these catalysts affected only the support; namely, part of $\beta$-$W_2C$ and $WO_2$ was transformed into $\alpha$-WC. For 10%Ni_10%Co/WC_DP and 20%Co/WC_DP (Table 1, entries 9 and 15), there was a transition of the cubic phases of nickel and cobalt to oxide and tungstate, respectively (Table 1, entries 20 and 22) during the DRM reaction. Additionally, for 10%Ni_10%Co/WC_DP, part of the tungsten carbide was oxidized by $CO_2$ to the oxide (Table 1, entry 20). In the case of 10%Ni_10%Co/WC_IWI (Table 1, entry 8) during the DRM reaction, $NiWO_4$ was converted to NiO and $WO_3$ to $WO_2$ (Table 1, entry 19). The 20%Co/WC_IWI material (Table 1, entry 15) turned to cobalt tungstate and graphite (Table 1, entry 21) during the DRM reaction.

The textural properties of the studied catalysts and support were characterized by a low-temperature adsorption-desorption of $N_2$, and the results are depicted in Table 1. It can be seen that the SSA values of samples prepared by DP varied in the range of 32.4–55.7 $m^2/g$ (Table 1, entries 3, 5, 7, 9, 11, 13 and 15) and were on average three to four times larger than the SSA values for catalysts prepared by IWI, in the range of 10.8–15.6 $m^2/g$ (Table 1, entries 2, 4, 6, 8, 10, 12 and 14), while the SSA of the support was only 4.5 $m^2/g$ (Table 1, entry 1). No direct correlation was observed between the SSA and the amounts of nickel or cobalt. The 5%Ni_15%Co/WC_DP and 10%Ni_10%Co/WC_DP materials showed the highest specific surface area among all catalysts (Table 1, entries 9 and 11).

Figure 1 shows TEM images, the size distribution of particles, EDX maps and SAED patterns of the studied materials. The presented data demonstrate that the samples prepared by DP (Figure 1, samples 3, 9 and 15) are characterized by a smaller particle size (14.9–20.3 nm) and distribution range (0–45 nm) than the samples prepared by IWI (22.8–33.3 nm and 0–160 nm, respectively) (Figure 1, samples 2, 8 and 14). The more uniform distribution of particles and its smaller size of materials prepared by the DP method are also confirmed by elemental mapping and electron diffraction patterns (the diffraction rings turn from continuous to point-like as the grain size of the polycrystalline material increases). Moreover, these data explain the difference in the specific surface area between IWI- and DP-prepared materials (Table 1), as a smaller particle size and more uniform distribution lead to a larger SSA. SAED (Figure 1, Table S1) and XRD (Table 1, entries 2, 3, 8, 9, 14 and 15) data are in full agreement. Based on the analysis of SAED patterns (Table S1), the following phases are identified: for sample 2—$WO_2$ (024) and (310), $\alpha$-WC (201); for sample 3—Ni (220), $\beta$-$W_2C$ (201), (203), (300) and (102), $\alpha$-WC (100), (101) and (111); for sample 8—$\alpha$-WC (111), $WO_3$ (143), (150), (2–42), (5–32), (−422) and (−504), $NiWO_4$ (014), $CoWO_4$ (−223), (242) and (410); for sample 9—$\beta$-$W_2C$ (102), (201), (302), (303) and (401), $\alpha$-WC (101) and (111); for sample 14—$CoWO_4$ (030), (032), (133) and (241), $\alpha$-WC (110); for

sample 15—Co (111), β-W$_2$C (002), (102), (110), (112), (201) and (301), α-WC (100), (111) and (201).

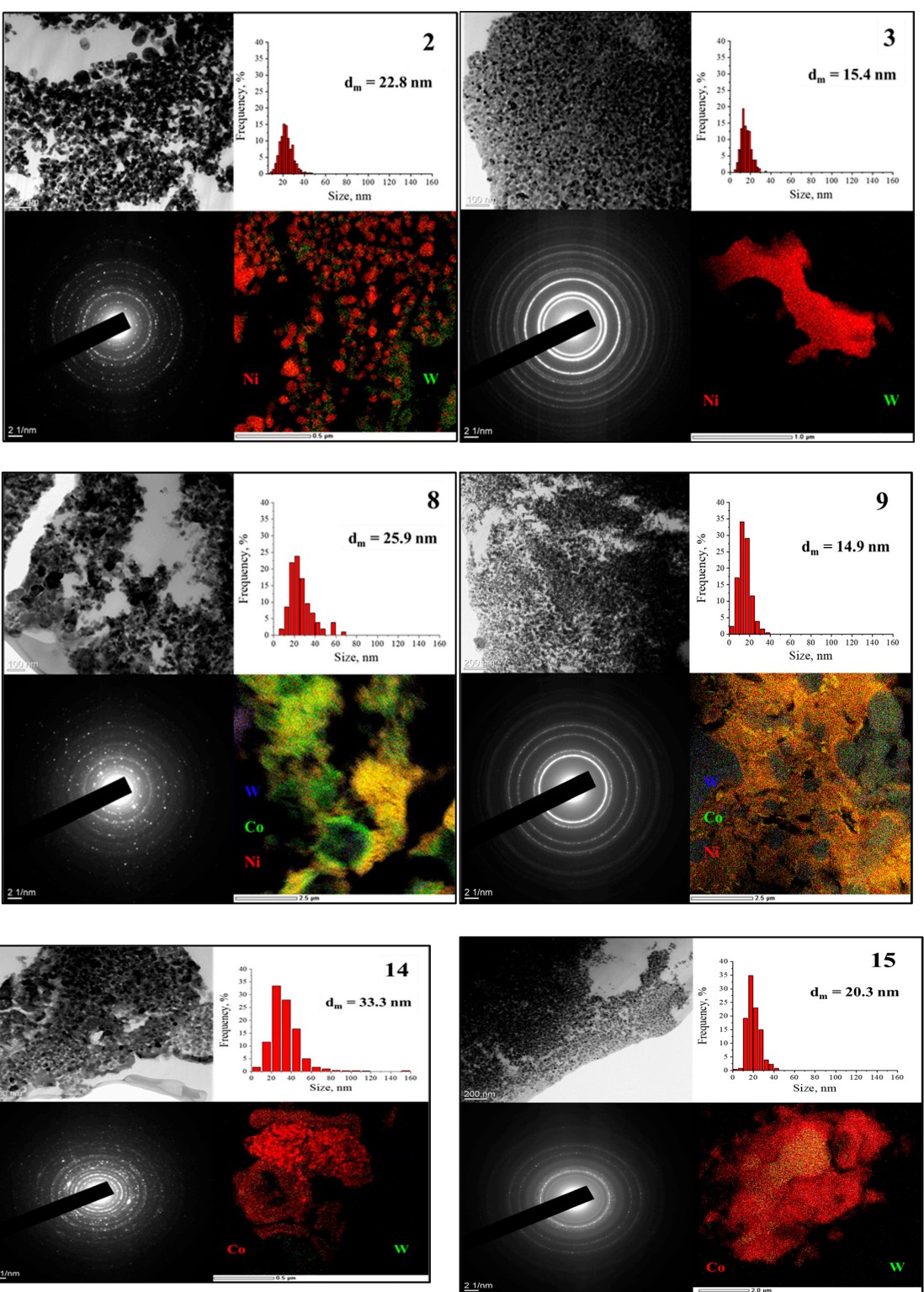

**Figure 1.** TEM images, particle distribution histograms, selected area electron diffraction (SAED) patterns and EDX elemental mapping of: (**2**)—20%Ni/WC_IWI; (**3**)—20%Ni/WC_DP; (**8**)—10%-Ni_10%Co/WC_IWI; (**9**)—10%Ni_10%Co/WC_DP; (**14**)—20%Co/WC_IWI; (**15**)—20%Co/WC_DP.

It is well known that the electronic state of the active metal plays an important role in DRM; in particular, the dissociation of the $CH_4$ molecules occurs at the metal centers, and, accordingly, the ratio between the metallic and ionic states of the metal is reflected in the catalytic behavior of the materials. In order to understand how the electronic state of nickel and cobalt changes on the support surface, depending on the preparation method and/or composition of the catalyst, the studied materials were analyzed by XPS (Table S2, Figures 2 and S3). As the nickel content in catalysts decreased from 20 to 1 wt.%, for both preparation methods, the amount of nickel in the metallic state decreases with a simultaneous increase in nickel in the oxidized state. At the same time, for samples prepared by DP (Table S2, Figure 2, samples 3, 9, 15 and Figure S3 samples 5, 7, 11), the amount of oxidized nickel increases on average by 1.2–1.5 times with a decrease in Ni content (from 20% for the sample with 20 wt.% Ni to 49% for the sample with 5 wt.% Ni), except for 1%Ni_19%Co/WC_DP (Table S2, Figure S3, sample 13), in which all nickel is in the oxidized state. For samples prepared by IWI (Table S2, Figure 2, samples 2, 8, 14 and Figure S3, samples 4,6), the amount of oxidized nickel increases by a factor of 2.1 with a decrease in nickel content (from 14% for the sample with 20 wt.% Ni to 58% for the sample with 10 wt.% Ni), while in samples containing 5 and 1 wt.% of Ni, all nickel is in the oxidized state (Figure S3, samples 10 and 12). In the case of cobalt, a decrease in its content from 20 to 10 wt.%, in samples prepared by DP, leads to an increase in the amount of Co in the oxidized states from 56 to 81% (Table S2, Figure 2, samples 9, 15 and Figure S3, samples 11,13); for samples with a cobalt content of 5 and 1 wt.%, the amount of the oxidized states decreases to 62 and 56%, respectively (Table S2, Figure S3, samples 7 and 5, respectively). For samples prepared by IWI with a cobalt content in the range of 1–20 wt.%, the amounts of Co in the oxidized and metallic states change slightly from 80 to 86% and from 20 to 14%, respectively (Table S2, Figure 2, samples 8, 14 and Figure S3, samples 4, 6, 10, 12).

## 2.2. Catalytic Results

The catalytic behavior of monometallic Ni and Co supported on tungsten carbide prepared by IWI or DP methods, and their bimetallic mixtures, was studied in dry methane reforming at 600–800 °C and a weight hourly space velocity (WHSV) of 3600–12,000 mL/g/h (Table S3). Based on the obtained data, the following general patterns can be identified: the conversion of methane and carbon dioxide increases with an increasing temperature, except for 5%Ni_15%Co/WC_IWI, 1%Ni_19%Co/WC_IWI and 20%Co/WC_IWI materials, and decreases with increasing a WHSV. In addition, the conversion of $CH_4$ and $CO_2$ is higher in the case of samples prepared by DP. It should also be noted that, for almost all studied materials, the $CO_2$ conversion is higher than the conversion of $CH_4$, and the $H_2/CO$ ratio is less than 1, which may be due to the following reactions [25]:

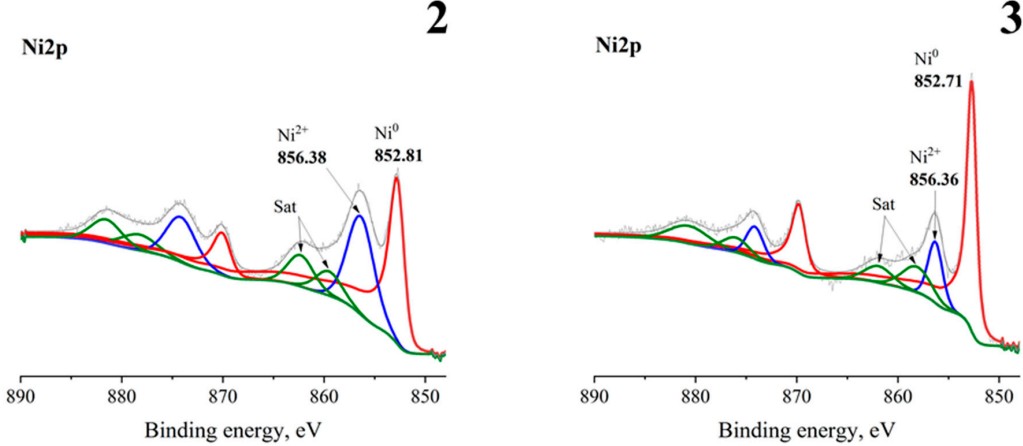

**Figure 2.** *Cont.*

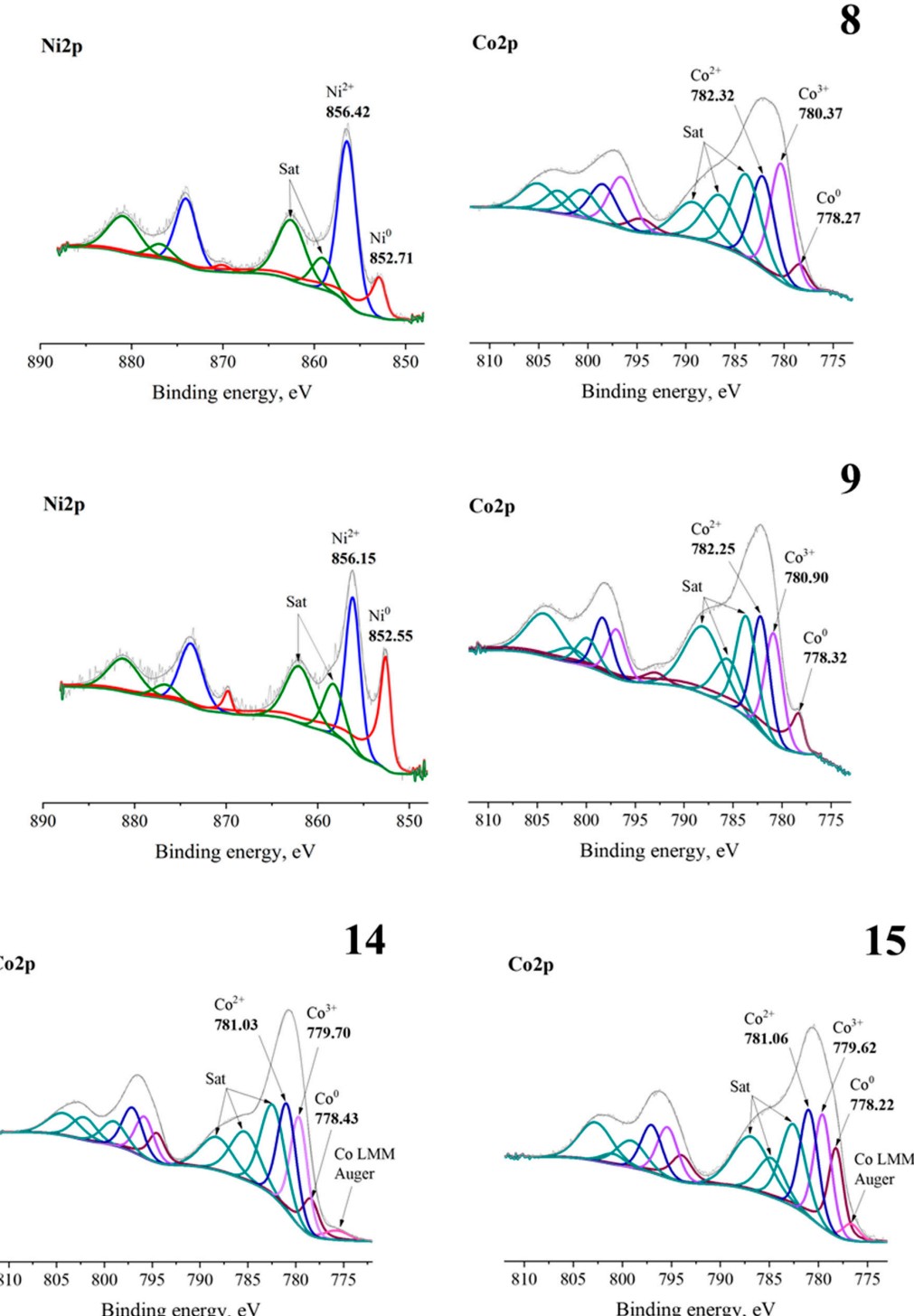

**Figure 2.** Ni2p XPS spectra: (**2**)—20%Ni/WC_IWI and (**3**)—20%Ni/WC_DP; Ni2p and Co2p XPS spectra: (**8**)—10%Ni_10%Co/WC_IWI and (**9**)—10%Ni_10%Co/WC_DP; Co2p XPS spectra: (**14**)—20%-Co/WC_IWI and (**15**)—20%Co/WC_DP.

1. The reverse Boudoir reaction, in this case, $CO_2$ from the feed, can gasify the carbon present on the catalyst surface:

$$CO_2 + C \leftrightarrow 2CO \tag{3}$$

2. The $CO_2$ activation on WC ($W_2C$) leads to oxidation of carbides by activated oxygen (O*):

$$CO_2 \rightarrow CO + O* \tag{4}$$

$$W_2C + 5O* \rightarrow 2WO_2 + CO \tag{5}$$

$$WC + 3O* \rightarrow WO_2 + CO \tag{6}$$

3. For the reverse water-gas-shift (RWGS), in this case, part of the $H_2$ produced is consumed by a reaction with $CO_2$:

$$CO_2 + H_2 \rightarrow CO + H_2O \tag{7}$$

Separately, it should be noted that one of the reasons for the decrease in the $CH_4$ and $CO_2$ conversion, with an increasing temperature, for 5%Ni_15%Co/WC_IWI, 1%Ni_19%-Co/WC_IWI and 20%Co/WC_IWI catalysts, may be the formation of nickel and/or cobalt tungstates under the action of the reaction medium and temperature (500–850 °C), which are not active in DRM below 850 °C [28], as well as the bulk oxidation of tungsten carbide [16]. At the same time, according to XRD data (Table 1, entries 10, 12, 14), $NiWO_4$ and/or $CoWO_4$, as well as $WO_3$ phases, were detected for these samples even before DRM. However, XPS data indicate that, at the surface of these samples, not only the oxidized states of cobalt were present but also its metallic state (Table S2, Figure 2, sample 14 and Figure S3, samples 10, 12). It is likely that this metal state is unstable and oxidizes during DRM, which leads to a decrease in the activity of these catalysts with an increasing temperature.

Figure 3 shows the dependences of $CH_4$ and $CO_2$ conversion as well as the $H_2/CO$ ratio at 800 °C and a WHSV of 3600–12,000 mL/g/h on the catalyst composition and preparation method. For materials prepared by DP, $CH_4$ and $CO_2$, the conversion decreases with an increasing cobalt content from 1 to 10 wt.%, and it further changes slightly from 10 to 15 wt.% Co; then, there is a little increase for 1%Ni_19%Co/WC_DP and finally a slight decrease for 20%Co/WC (Figure 3a,c). Increasing a WHSV from 3600 to 12,000 mL/g/h reduces both the $CH_4$ and $CO_2$ conversion. The $H_2/CO$ ratio, as in the case of the $CH_4$ and $CO_2$ conversion, decreases with an increase in the cobalt content up to 10 wt.%; a further increase in the cobalt amount from 10 to 20 wt.% does not lead to significant changes in the $H_2/CO$ ratio (Figure 3e). An increase in the WHSV leads to a decrease in the $H_2/CO$ ratio. For materials prepared by IWI, $CH_4$ and $CO_2$, the conversion decreases with an increasing cobalt content from 1 to 20 wt.% (Figure 3b,d); an increase in the WHSV also leads to a decrease in the conversion of both $CH_4$ and $CO_2$. The $H_2/CO$ ratio decreases from 0.56 to 0.40 for a WHSV of 3600 mL/g/h and from 0.55 to 0.33 for a WHSV of 6000 mL/g/h with an increasing cobalt content in the catalysts (Figure 3f). For a WHSV of 12,000 mL/g/h, the $H_2/CO$ ratio varies within 0.38–0.41, except for samples containing 19 and 20 wt.% of Co, for which, due to the low conversion of reagents, the $H_2/CO$ ratio is 0.50 and 1.00, respectively.

Thus, comparing the catalytic and physico-chemical data, it can be concluded that the higher catalytic performance of materials prepared by DP is due to the combination of their structural, textural and electronic properties (Table 1, entries 3, 5, 7, 9, 11, 13 and 15, Table S1, samples 3, 9 and 15, Table S2, samples 3, 5, 7, 9, 11, 13 and 15, Figures 1 and 2, samples 3, 9 and 15, Figure S1, samples 3, 5, 7, 9, 11, 13 and 15, Figure S2, samples 5, 7, 11 and 13). At the same time, a change in the composition of DP-prepared catalysts, namely, an increase in the cobalt content along with a decrease in the nickel amount, leads to a decrease in the catalytic efficiency (a decrease in the $CH_4$ and $CO_2$ conversion and $H_2/CO$ ratio, Figure 3a,c,e). Thereby, the textural and structural characteristics do not change significantly, but the alterations are clearly seen in the modification of the electronic properties of these materials, in particular, an increase in the amount of oxidized states of Ni and/or Co along with a decrease in their metallic states. That is, before DRM, the differences are visible only on the surface layer of these catalysts, while after DRM, changes are also detected in the phase composition (Table 1, entries 18, 20 and 22, Figure S2, samples

18, 20 and 22). Accordingly, it can be assumed that the transition from the metallic states of nickel and/or cobalt to their oxidized states, as well as of tungsten carbide to its oxide, is the reason for the deactivation of catalysts in DRM. In this case, the most stable and active DRM catalyst, among others studied, is 20%Ni/WC_DP.

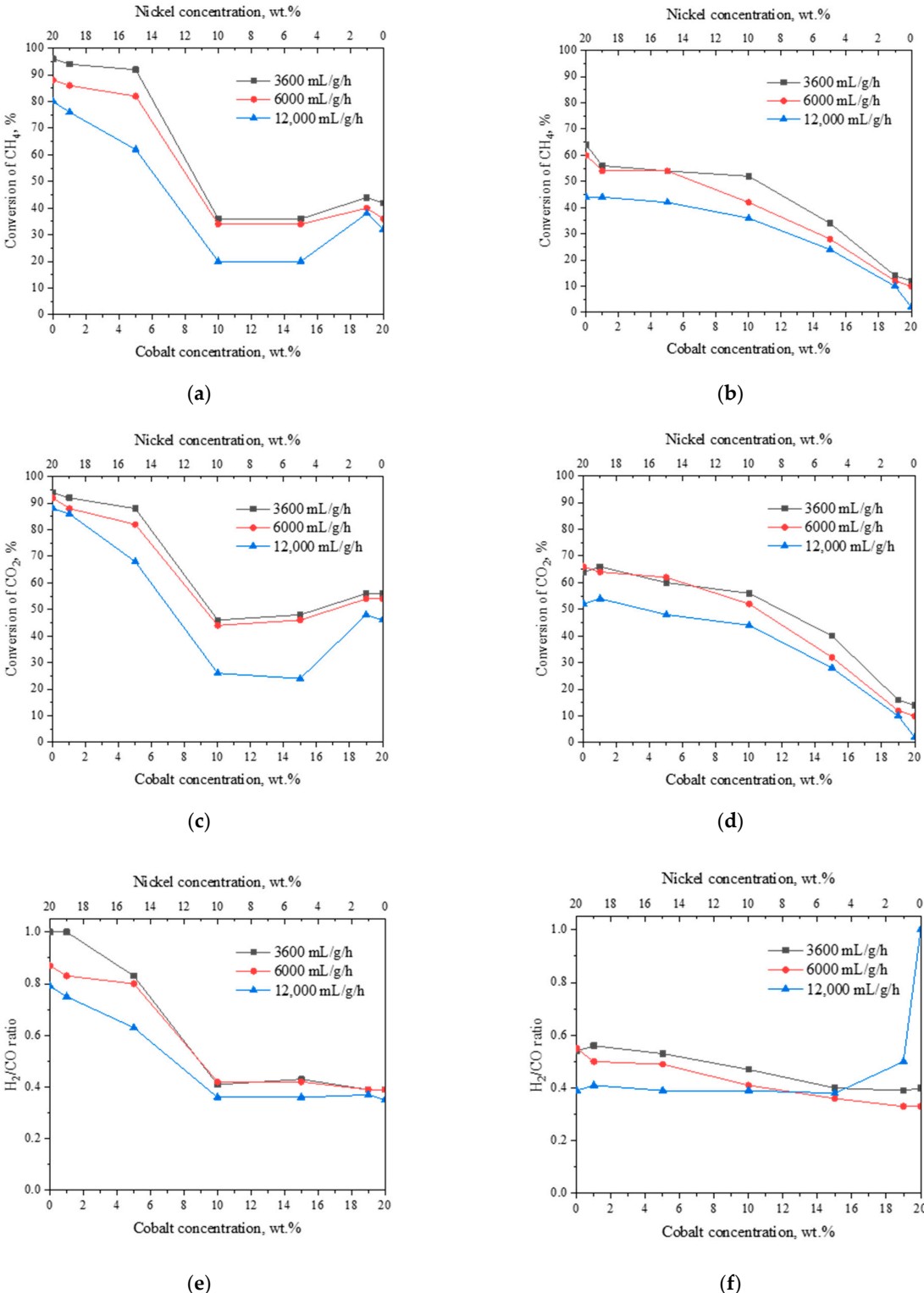

**Figure 3.** DRM performance at various catalyst compositions and WHSV for samples prepared by DP (**a,c,e**) and IWI (**b,d,f**) methods (T = 800 °C, P = 1 atm, $CO_2/CH_4$ = 1).

The results of the stability study of 20%Ni/WC_DP in DRM at 800 °C and a WHSV of 12,000 mL/g/h are shown in Figure 4. The conversion of $CO_2$ and $CH_4$ (Figure 4a), as well as the $H_2/CO$ ratio (Figure 4b), after 1 h and 80 h of time on stream (TOS) is 88 and 84%, 80 and 72%, 0.79 and 0.74, respectively. Quantitatively, the catalyst deactivation degree after 80 h of TOS based on the $CH_4$ and $CO_2$ conversion can be estimated as follows [31]:

$$D = \frac{X_{CH_4,2h} - X_{CH_4,80h}}{X_{CH_4,80h}} \times 100\% \tag{8}$$

$$D = \frac{X_{CO_2,1h} - X_{CO_2,80h}}{X_{CO_2,80h}} \times 100\% \tag{9}$$

where X is the $CH_4$ or $CO_2$ conversion after 1 h and 80 h TOS. As a result, the degree of catalyst deactivation is 10 and 5% for $CH_4$ and $CO_2$, respectively. This may mean that the carbon formation on the catalyst surface is more intense than its removal by the reverse Boudoir reaction (Equation (3)) or, as suggested in [25], through the oxidation/(re)carburization cycle, the first stage of which is described by Equations (2) and (4)–(6), followed by the (re)carburization stage of tungsten oxide by pyrolytic carbon (C*):

$$2WO_2 + 5C^* \rightarrow W_2C + 4CO \tag{10}$$

$$WO_2 + 3C^* \rightarrow WC + 2CO \tag{11}$$

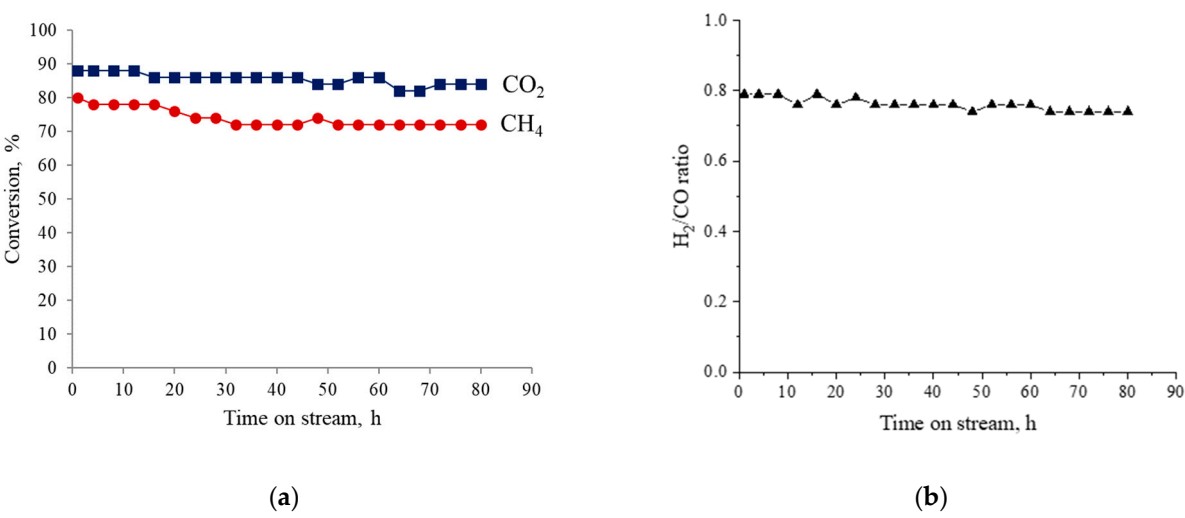

(**a**)    (**b**)

**Figure 4.** (**a**) $CH_4$ and $CO_2$ conversion; (**b**) $H_2/CO$ ratio as a function of time on stream over 20%Ni/WC_DP catalyst in DRM (T = 800 °C, P = 1 atm, $CH_4/CO_2$ = 1 and WHSV = 12,000 mL/g/h).

One of the options to equalize the rates of carbon formation and its removal from the catalyst surface can be the use of $CO_2$ excess, promoting effective removal of carbon from the catalyst surface, according to Equations (4)–(6), (10) and (11).

Finally, the efficiency of the catalyst synthesized in the current study was evaluated by comparing its catalytic performance for DRM with the results obtained by other authors from the literature (Table 2).

The presented results illustrate that the 20%Ni/WC_DP material shows an activity performance comparable with (and, in some cases, superior to) the catalysts found in the literature for DRM. In addition, as shown above (Table 1, entry 18, Figure S2, sample 18 and Figure 4a), 20%Ni/WC_DP is stable in the DRM conditions. Moreover, its catalytic performance can be improved by varying the feed composition; another option may be to modify its composition and/or textural properties.

**Table 2.** Ni-WC, Co-WC and WC catalysts used in DRM.

| Entry | Sample | Reaction Conditions | | | Conversion, % | | $H_2$/CO | Ref. |
|---|---|---|---|---|---|---|---|---|
| | | **P, atm** | **T, °C** | **WHSV, mL/h/$g_{cat}$** | **CO$_2$** | **CH$_4$** | | |
| 1 | 20%Ni/WC_DP | 1 | 800 | 3600<br>6000<br>12,000 | 94<br>92<br>88 | 96<br>88<br>80 | 1.00<br>0.87<br>0.79 | This work |
| 2 | Ni-WC$_x$ | 1 | 800 | 18,000 | 71 | 58 | 0.69 | [19] |
| 3 | Co$_6$W$_6$C | 5 | 850 | 11,200 | 78 | 82 | 1.01 | [20] |
| 4 | Ni–W–C | n.d. | 850 | n.d. | 77 | 78 | 0.99 | [21] |
| 5 | Co–W–C | 3.4 | 850 | 9000 | 74 | 79 | 1.11 | [21] |
| 6 | Ni-WC | 1 | 800 | n.d. | 85 | 75 | 0.79 | [22] |
| 7 | Co-$\beta$W$_2$C/$\alpha$WC | 1 | 800 | 36,000–72,000 | 90 | 82 | 0.86 | [23] |
| 8 | WC | 5 | 850 | 4000 [a] | 100 | 78 | 0.85 | [24] |
| 9 | WC | n.d. | 970 | n.d. | 83 | 62 | 0.81 | [25] |
| 10 | WC | 0.867 | 843 | n.d. | n.d. | 34 | 1.22 | [26] |

[a] **GHSV (1/h)**.

## 3. Materials and Methods

### 3.1. Synthesis of Support

The synthesis of tungsten carbide was performed on a laboratory electric arc unit, which allows the effect of self-shielding of the reaction volume from air oxygen during plasma treatment of the starting material. The laboratory electric arc installation includes an inverter to which electrodes are connected: a cathode and an anode. The cathode is a graphite crucible, and the anode is a graphite rod. The anode is mounted above the cathode with the possibility of movement by means of a linear electric drive with a stepper motor. The cathode is fixed with current-carrying clamps; an initial mixture of carbon and tungsten is poured in an even layer on the bottom of the cathode, and a pad of graphite felt is placed over the mixture and subjected to an arc discharge. Commercial carbon and tungsten powders (purity not less than 99.9 wt.%) with an average particle size of not more than 5–7 μm were used. The operating current of the power supply was 220 A at an open circuit voltage of 63 V. The values of current strength and duration of the arc discharge maintenance were chosen on the basis of previously obtained data [32]. Separately, it should be noted that the advantages of this method are a short synthesis time of 45 s, an economy of precursors, since only carbon and tungsten are used. Moreover, in this method, there is no need for vacuum equipment, which reduces energy consumption.

### 3.2. Synthesis of Catalysts

The catalysts named as 20%Ni/WC_DP, 19%Ni_1%Co/WC_DP, 15%Ni_5%Co/ WC_DP, 10%Ni_10%Co/WC_DP, 5%Ni_15%Co/WC_DP, 1%Ni_19%Co/WC_DP and 20%Co/WC_DP were prepared by deposition-precipitation with NaOH (DP). To an aqueous solution containing the calculated amounts of Ni(NO$_3$)$_2$·6H$_2$O or/and Co(NO$_3$)$_2$·6H$_2$O, 0.8 g of the support was added, followed by heating to 80 °C. The pH value, initially around 3, was adjusted to 9 by the dropwise addition of the 0.5 M NaOH solution. The suspension was kept at 80 °C for 2 h under vigorous stirring. Further, precipitates were washed and centrifuged several times, and then dried in a vacuum for 2 h at 100 °C.

The catalysts named as 20%Ni/WC_IWI, 19%Ni_1%Co/WC_IWI, 15%Ni_5%Co/ WC_IWI, 10%Ni_10%Co/WC_IWI, 5%Ni_15%Co/WC_IWI, 1%Ni_19%Co/WC_IWI and 20%Co/WC_IWI were prepared by incipient wetness impregnation (IWI) of the support (0.8 g) with an aqueous solution containing the desired amounts of Ni(NO$_3$)$_2$·6H$_2$O or/and

$Co(NO_3)_2 \cdot 6H_2O$. After 2 h of impregnation at room temperature, the samples were calcined at 600 °C for 3 h.

The nominal nickel or cobalt content in the monometallic catalysts was 20 wt.%; in the bimetallic ones, it varied in the range of 1–19 wt.% (i.e., the total content of nickel and cobalt in the sample was 20 wt.%, and the content of each varied from 1 to 19 wt.%).

After nickel and/or cobalt deposition on the support surface and the drying/calcination procedure, all obtained materials were pretreated in $H_2$ at 600 °C for 2 h.

### 3.3. Characterization of Support and Catalysts

The phase composition of the support and catalysts was identified by X-ray diffraction (XRD) using a Shimadzu XRD 7000s X-ray diffractometer (Shimadzu XRD 7000s, $\lambda = 1.5406$ Å, step of 0.02 deg, exposure of 1 s per point, X-ray tube voltage of 40 kV, current of 30 mA). Qualitative analysis was carried out using the international structural database PDF4+. Quantitative analysis was carried out by the Rietveld method and the Reference Intensity Ratio (RIR) methods. The RIR method is based upon scaling all diffraction data to the diffraction of standard reference materials. The intensity of a diffraction peak profile is a convolution of many factors, only one of which is the concentration of the analyte (species being measured). By using the RIR method, ratios scaled to a common reference are used in the experiment. The assumption is that all the factors, except the concentration of the analyte, are ratioed and reduced to a constant. By using ratios and measuring peak areas, the RIR method can be used to determine concentrations. The Rietveld method uses a least squares approach to refine a theoretical line profile until it matches the measured profile. Both methods give similar results accurate to within ± 3 wt.% at the 95% confidence level. The calculation error strongly depends on the studied material, and in the current study it is 2–3%.

The textural properties of the support and catalysts were studied on a Sync220A instrument (3P Instruments GmbH & Co. KG, Odelzhausen, Germany). Before measurements, the materials were degassed in a vacuum at 130 °C for 12 h. The Brunauer–Emmett–Teller (BET) method was used to calculate the specific surface area from the adsorption isotherm in the relative pressure range from 0.005 to 0.25. The pore size distribution was calculated from the desorption data using the Barrett–Joyner–Halenda (BJH) method.

Energy-dispersive X-ray spectroscopy (EDX), transmission electron microscopy (TEM) in conjunction with selected area electron diffraction (SAED) were performed using a JEM-2100F microscope (JEOL Ltd., Tokyo, Japan). Before the TEM studies, samples were prepared by ion thinning using an Ion Slicer EM-09100 IS (JEOL). Preliminary sample preparation for the Ion Slicer consisted in the manufacture of 2 plates (lamellas) with dimensions of 4.5 mm × 0.50 mm × 0.15 mm, and gluing their ends (from the thin wide end) using a mixture of epoxy resin and the powder of the studied material. In the Ion Slicer, the prepared sample was closed from the thin wide end with a special protective tape and thinned with an argon ion beam. The beam energy did not exceed 8 kV, and the beam tilt angle was 6°, with respect to the largest face of the sample. This made it possible to minimize radiation damage and thereby preserve the original structure and phase composition of the samples. The interpretation of the SAED patterns was based on matching the d-spacing from SAED patterns, obtained in this study, with the d-spacing values from the PDF4+ international structural database (Table S1).

X-ray photoelectron spectroscopy (XPS) was used to study the elemental composition of catalysts, the chemical/electronic state of Ni and Co on the surface layer (3–15 nm) of the studied catalysts. XPS measurements were performed using a Thermo Fisher Scientific XPS NEXSA spectrometer with a monochromated Al K Alpha X-ray source working at 1486.6 eV. The survey spectra and high-resolution spectra of individual elements were recorded at pass energy 200 and 50 eV with a step of 1 and 0.1 eV, respectively. The flood gun was used for the charge compensation. Additionally, if necessary, the charge effect was corrected using the C1s peak at 285.0 eV as a reference. The obtained spectra were processed using Avantage Thermo Fisher and CasaXPS software (version 2.3.15, CASA

Software Ltd., Teignmouth, UK, http://www.casaxps.com/). It is worth noting that the interpretation of the chemical state of nickel and cobalt on the surface of the studied catalysts is a difficult task, especially if both metallic and ionic states of the element are present due to the complexity of their 2p spectra caused by peak asymmetries, complex multiplet splitting, shake-up and plasmon loss features. The deconvolution of the spectra and the interpretation of the electronic state of nickel and cobalt were carried out on the basis of the literature data [26,33–45].

### 3.4. Catalytic Experiments

The catalytic properties of the synthesized samples were studied in the dry reforming of methane at the temperature range of 600–800 °C, weight hourly space velocity (WHSV) of 3600–12,000 mL/g/h and atmospheric pressure in a quartz flow reactor, with a fixed bed of the catalyst, using a gas mixture of 50 vol.% $CO_2$ and 50 vol.% $CH_4$ (flow rate 30–100 mL/min). The effluent gas composition was monitored every hour by an online gas chromatograph (CHROMOS GC-1000, CHROMOS, Dzerzhinsk, Russia) equipped with two TCD detectors and two separate packed columns filled with CaA (to analyze $O_2$ and $H_2$) and AG-3 sorbent (to analyze $CH_4$, $CO_2$ and CO), and He as the carrier gas. The concentrations of reactants and products were determined by the absolute calibration method. Each measurement was carried out twice.

The catalytic performance was evaluated by the magnitude of $CH_4$ and $CO_2$ conversions (X%) and the $H_2$/CO ratio as in the following equations:

$$X_{CO_2} = \frac{[CO_2]_{in} - [CO_2]_{out}}{[CO_2]_{in}} \tag{12}$$

$$X_{CH_4} = \frac{[CH_4]_{in} - [CH_4]_{out}}{[CH_4]_{in}} \tag{13}$$

$$\frac{H_2}{CO} = \frac{[H_2]_{out}}{[CO]_{out}} \tag{14}$$

where $[i]_{in}$ and $[i]_{out}$ are the concentration of each component in the inlet and outlet feed, respectively; i is $CO_2$, $CH_4$, $H_2$ or CO.

The time on stream (TOS) experiment was carried out at 800 °C and a WHSV of 12,000 mL/g/h.

## 4. Conclusions

Catalysts of nickel, cobalt and nickel-cobalt were obtained by DP and IWI preparation methods, using tungsten carbide produced by the vacuum-free electric arc method as the support. The resulting materials were tested for DRM and characterized by $N_2$ adsorption-desorption, TEM in conjunction with SAED and EDX, XRD and XPS methods. The DP-prepared catalysts were the most efficient in DRM due to the combination of their structural, textural, and electronic properties. However, the catalytic performance and stability of these materials were sensitive to changes in their composition. An increase in the cobalt content along with a decrease in the nickel amount led not only to a change in their electronic properties, which was traced even before DRM, but also to a change in their bulk structure after DRM. The most efficient and stable catalyst among all those studied was 20%Ni/WC_DP, which showed the best resistance to oxidation and coking during DRM. Its degree of deactivation, with respect to methane and carbon dioxide for 80 h TOS, was 10 and 5%, respectively. These results lead to the conclusion that the rate of carbon deposition is higher than the rate of its removal from the catalyst surface by the reverse Boudoir reaction and/or oxidation/(re)carbonization cycle. It is assumed that the carbon removal rate can be increased by using an excess of $CO_2$ or by modification of the catalyst composition. This will be the focus of our subsequent research.

**Supplementary Materials:** The following are available online at https://www.mdpi.com/article/10 .3390/catal12121631/s1, Figure S1: XRD patterns of support and catalysts vs. XRD patterns of the reference phases: (1)—WC, (2)—20%Ni/WC_IWI, (3)—20%Ni/WC_DP, (4)—19%Ni_1%Co/WC_IWI, (5)—19%Ni_1%Co/WC_DP, (6)—15%Ni_5%Co/WC_IWI, (7)—15%Ni_5%Co/WC_DP, (8)—10%-Ni_10%Co/WC_IWI, (9)—10%Ni_10%Co/WC_DP, (10)—5%Ni_15%Co/WC_IWI, (11)—5%Ni_15%-Co/WC_DP, (12)—1%Ni_19%Co/WC_IWI, (13)—1%Ni_19%Co/WC_DP, (14)—20%Co/WC_IWI and (15)—20%Co/WC_DP, Figure S2: XRD patterns of spent (used) support and catalysts vs. XRD patterns of the reference phases: (16)—WC, (17)—20%Ni/WC_IWI, (18)—20%Ni/WC_DP, (19)—10%- Ni_10%Co/WC_IWI, (20)—10%Ni_10%Co/WC_DP, (21)—20%Co/WC_IWI and (22)—20%Co/WC_DP, Figure S3: Ni2p and Co2p XPS spectra of (4)—19%Ni_1%Co/WC_IWI, (5)—19%-Ni_1%Co/WC_DP, (6)—15%Ni_5%Co/WC_IWI, (7)—15%Ni_5%Co/WC_DP, (10)—5%Ni_15%-Co/WC_IWI, (11)—5%Ni_15%Co/WC_DP, (12)—1%Ni_19%Co/WC_IWI and (13)—1%Ni_19%-Co/WC_DP, Table S1: Lattice d-spacing of studied materials, determined from SAED pattern, and its corresponding phases vs. references values, Table S2: The relative content of different Ni and Co electronic states on the surface of the studied samples, calculated according to XPS, Table S3: Catalytic behavior of studied materials in methane dry reforming.

**Author Contributions:** A.P. (Alexander Pak) and E.K.: conceptualization; A.P. (Alexander Pak) and E.K.: methodology; Z.B., D.G., E.P. and E.K.: software; Z.B., D.G., E.P., K.L. and E.K.: investigation; Z.B., D.G., E.P. and E.K.: visualization; A.P. (Alexey Pestryakov), A.P. (Alexander Pak), S.A.C.C., N.B. and E.K.: data curation; Z.B., D.G., E.P. and E.K.: writing—original draft preparation; A.P. (Alexey Pestryakov), A.P. (Alexander Pak), K.L., S.A.C.C., N.B. and E.K.: writing—review and editing; E.K., A.P. (Alexey Pestryakov) and A.P. (Alexander Pak): supervision; A.P. (Alexey Pestryakov), A.P. (Alexander Pak), N.B. and S.A.C.C.: resources; A.P. (Alexey Pestryakov): project administration; A.P. (Alexey Pestryakov): funding acquisition. All authors have read and agreed to the published version of the manuscript.

**Funding:** This research was funded by Tomsk Polytechnic University grant Priority-2030-NIP/EB-045-1308-2022. This work was also supported by FCT/MCTES (Fundação para a Ciência e Tecnologia and Ministério da Ciência, Tecnologia e Ensino Superior) through projects UIDB/50006/2020 and UIDP/50006/2020. S.A.C.C. also acknowledges FCT for the Scientific Employment Stimulus—Institutional Call (CEECINST/00102/2018).

**Data Availability Statement:** Data available upon request.

**Acknowledgments:** XPS measurements were carried out at the Central laboratories of Tomsk Polytechnic University (Analytical Center). TEM, including EDX and SAED experiments, was carried out using the equipment of the Center for Sharing Use "Nanomaterials and Nanotechnologies" of Tomsk Polytechnic University supported by the RF Ministry of Education and Science project #075-15-2021-710. Research on the synthesis of tungsten carbide was supported by the RF Ministry of Education and Science project No FSWW-2022-0018.

**Conflicts of Interest:** The authors declare no conflict of interest.

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
