# Peer review of "Ni, Co and Ni-Co-Modified Tungsten Carbides Obtained by an Electric Arc Method as Dry Reforming Catalysts"

_catalysts, doi:10.3390/catal12121631_

Round 1
Reviewer 1 Report
The manuscript is clear and generally well written, the data look interesting. However, the authors draw unfounded conclusions, which does not allow me to recommend the publication of the article. Major revision is needed. The main drawbacks are listed below.
1) I recommend to replace “like Rh, Ru, Pd, Pt, and Ir” by “such as Rh, Ru, Pd, Pt, and Ir”, “X-ray diffractograms” by “XRD patterns”, “the specific surface area (SBET)” by “the specific surface area (SSA)”, and “the SBET” by “SSA”. Abbreviations (DP, IWI) need to be entered only once.
2) The presented XRD data do not allow me to verify the correctness of the phase analysis performed. The authors presented 15 diffractograms in one figure (Fig. S1). Since there are no limits on the size of the supplementary materials, it is possible to depict these data in more detail in several figures. I also recommend to show bar diagrams of the reference phases in the figures.
3) Table 1 provides quantitative information on the phase composition of the catalysts under study, but the authors do not describe how and with what accuracy these data were obtained.
4) The analysis of SAED patterns is also inconclusive and not supported by references.
5) The authors provide a description of the conclusions obtained from the XPS data analysis (page 7), but it is not clear on the basis of what these conclusions were made. It is necessary to indicate the observed values of the binding energies and describe the shape of the spectra, then, based on a comparison with the literature data, draw a conclusion about the chemical state of the elements.
6) The presentation of the Ni2p spectra is very nice. However, it is not clear why the satellite structure substantially depends on the method of preparation and the nickel loading. This means that the chemical state of nickel in the catalysts is different and can affect their catalytic performance.
7) The shape of the Co2p core-level spectra is unusual and cannot be described by the presence of cobalt in two chemical states. Obviously, not only metallic cobalt is present on the surface of the catalysts, but also oxidized cobalt in two different states.
8) The authors write about “chemical/electronic state of Ni and Co”. However, there is no data in the text on how the electronic state of metals or oxides changes.
Author Response
Please, find an attached file

Reviewer 2 Report
11/20/2022
Review on Manuscript ID: catalysts-2063732
Authors investigated a series of Ni, Co and Ni-Co-modified tungsten carbides catalyst for CH4 dry reforming. One of unique of the study is authors used vacuum-free electric arc to prepare WC support, which is different from the conventional precipitation method. Authors study effect of cobalt and Nickel contents on the physical properties including electronic and bulk structure and catalytic performance of WC supported catalysts prepared by IWI and DP methods. Various techniques such as SAED and EDX, XRD and XPS were used to understand the relationship between catalyst structure and reaction performance; Catalyst oxidation and carburization cycle model is proposed and supported by reaction and the characterization data. This model can be used to explain higher CO2 conversion than CH4 conversion authors achieved do, and it also can explain catalyst deactivation.
More importantly, author developed 20%Ni/WC_DP catalyst (index 3 in table 1) that stand out among a series of WC catalysts (22) in MDR reaction, it shows the superior catalyst activity (>90% Conversion) and good oxidation resistance, thus good stability (80 h) at 800 oC.
This is a good paper containing informative data, it is suitable for publishing in Catalysts.
Comments
Can authors address a little bit more the advantage of electric arc method relative conventional precipitation or DP method in the introduction or Catalyst preparation?
Figure 1 TEM results only show the entries 2, 3, 8, 9, 14, 15 prepared by DP and IWI, results of other samples are not shown in Supplementary Materials, while the XRD results of all 22 samples are given in Table 1? Are the TEM data of other samples support your conclusion DP method generated smaller particle size than IWI?
Author Response
Please, find an attached file

Reviewer 3 Report
The work is devoted to an important topic - the improvement of catalysts for the processing of carbon dioxide.
Used modern methods to describe the catalysts.
The article corresponds to the subject of the journal and can be published in this form.
A small mistake on 3 pages in 102 line (the degree sign is not required).
Author Response
Please, find an attached file

Reviewer 4 Report
Methane reforming is an important field of clean energy. This paper is a very good work on methane reforming. It provides a good research idea for related research in this field. However, the following problems exist:
1. The experimental method should be quoted reasonably.
2. Significant figures and experimental errors shall be correctly expressed.
3. The expression form of Figure 3 is a little confusing, and it is recommended to use the conventional expression form.
4. The author needs to supplement the XRD spectrum. Don't just use Table.
5. The mechanism analysis needs to be further strengthened.
Author Response
Please, find an attached file

Round 2
Reviewer 1 Report
After revision, the manuscript became better.